# Integrative Neoepitope Discovery in Glioblastoma via HLA Class I Profiling and AlphaFold2-Multimer

**DOI:** 10.3390/biomedicines13112715

**Published:** 2025-11-05

**Authors:** Raquel Francés, Jenny Bonifacio-Mundaca, Íñigo Casafont, Christophe Desterke, Jorge Mata-Garrido

**Affiliations:** 1Anatomy and Cell Biology Department, University of Cantabria-IDIVAL, 39011 Santander, Spain; rfrancesr@unican.es (R.F.); inigo.casafont@unican.es (Í.C.); 2National Tumor Bank, Department of Pathology, National Institute of Neoplastic Diseases, Surquillo 15038, Peru; jenny.bonifacio@upch.pe; 3Faculté de Médecine du Kremlin Bicêtre, Université Paris-Saclay, 94270 Le Kremlin-Bicêtre, France

**Keywords:** HLA-A*68:01, HLA-B*15:01, glioblastoma, Colabfold, mhcflurry2, neoantigens, immunopeptidomics

## Abstract

**Background/Objectives:** Glioblastoma multiforme (GBM) is an aggressive primary brain tumor with limited therapeutic options. Neoantigen-based immunotherapy offers a promising avenue, but its efficacy primarily depends on the ability of somatic mutations to generate immunogenic peptides effectively presented by HLA class I molecules and recognized by cytotoxic T cells, in concert with innate immune mechanisms such as NK-cell activation and DAMP/PAMP signaling. This study aimed to characterize the MHC-I binding diversity of peptides derived from GBM-associated somatic variants, with a particular focus on interactions involving HLA-A68:01 and HLA-B15:01 alleles. These alleles were selected based on their ethnic prevalence and potential structural compatibility with neoepitopes. **Methods:** Somatic missense variants from TCGA-GBM were filtered using high-confidence genomic databases, including dbSNP, COSMIC, and MANE. Neoepitope prediction was performed across multiple HLA class I alleles using binding affinity algorithms (MHCflurry2). Peptide–HLA interactions were characterized through motif analysis and anchor residue enrichment. Structural modeling of peptide–HLA complexes was conducted using ColabFold (AlphaFold2-multimer v3) to evaluate conformational stability. The population frequency of selected HLA alleles was examined through epidemiological comparisons. **Results:** Canonical GBM driver mutations (e.g., EGFR, TP53, PIK3R1) are recurrent and biologically relevant, although pharmacological inhibition of EGFR alone has not consistently improved patient outcomes, underscoring the complex signaling redundancy in glioblastoma. HLA-A68:01 exhibited high binding affinity and favorable motif compatibility, supporting its potential for effective neoantigen presentation. HLA-B15:01 was identified as a viable presenter for the EGFR p.Arg108Lys variant. Structural modeling confirmed stable peptide insertion into the MHC-I binding groove, with high-confidence folding and preserved interface integrity. Ethnic distribution analysis revealed varying GBM incidence across populations expressing these alleles. **Conclusions:** This integrative analysis identified structurally validated, immunogenically promising neoantigens derived from GBM mutations, particularly for HLA-A68:01 and HLA-B15:01. These findings support allele-informed neoepitope prioritization in personalized immunotherapy, especially for patient populations with corresponding HLA genotypes and MHC-I presentation capacity.

## 1. Introduction

According to the 2021 World Health Organization (WHO) Classification of Tumors of the Central Nervous System, glioblastoma is defined as an IDH-wildtype astrocytic tumor, CNS WHO grade 4, whereas astrocytic tumors that arise from lower-grade precursors and harbor IDH mutations are now classified as astrocytoma, IDH-mutant, CNS WHO grade 4. This updated nomenclature replaces the former distinction between “primary” and “secondary” glioblastoma, reflecting a shift toward molecularly based taxonomy rather than purely histopathological criteria [1,2].

Despite this nomenclature change, the biological differences that historically defined these two entities remain relevant for understanding tumor pathogenesis. Tumors now classified as glioblastoma (formerly “primary GBM”) typically present de novo and exhibit epidermal growth factor receptor (EGFR) amplification, PTEN mutations, and chromosome 10 loss [3,4,5]. In contrast, astrocytoma, IDH-mutant, grade 4 (formerly “secondary GBM”) usually evolves from lower-grade gliomas and is characterized by recurrent mutations in *IDH1* and *TP53*, often accompanied by 19q loss [3,4,5]. Importantly, *IDH1* mutations are consistently associated with improved survival outcomes [6].

GBM is an exceptionally heterogeneous and treatment-refractory malignancy marked by extensive spatial and temporal genetic diversity, a highly immunosuppressive tumor microenvironment, and frequent resistance to standard cytotoxic therapies. Molecular profiling has revealed multiple overlapping pathways driving GBM progression—including *EGFR* amplification/mutation, alterations in *TP53*, *PTEN* loss, and activation of PI3K/AKT signaling—that not only drive tumorigenesis but also shape antigenic landscapes and mechanisms of immune evasion. The poor prognosis of GBM patients despite aggressive therapy highlights the urgent need for new therapeutic modalities. Neoantigen-directed immunotherapies (personalized vaccines, adoptive T-cell therapies, or TCR-engineered approaches) represent a rational strategy because somatic mutations unique to tumor cells can generate peptides absent from normal tissues and therefore potentially elicit tumor-selective T-cell responses.

Recent reviews have underlined that the molecular complexity of GBM contributes not only to treatment resistance but also to the modulation of its electrical, optical, and metabolic properties, which may influence immunotherapy responses [7]. Likewise, drug-resistance mechanisms linking DNA repair, metabolic rewiring, and immune evasion have been comprehensively summarized [8]. Preclinical models, including patient-derived xenografts that preserve tumor heterogeneity and HLA expression, provide valuable systems for validating neoantigen discovery pipelines [9]. These studies highlight the multifaceted biology of GBM and the pressing need for integrative approaches that connect genomic, immunologic, and structural data.

Computational neoantigen discovery pipelines have become indispensable for prioritizing candidate neoepitopes from tumor sequencing data before committing resources to experimental validation. These pipelines typically integrate mutation annotation, peptide generation, antigen processing and MHC binding prediction, and increasingly, structural assessment of peptide–MHC to refine epitope prioritization. In GBM, however, the added challenges of heterogeneous mutational burdens, regional MHC-I downregulation, and an immunosuppressive microenvironment necessitate a conservative and integrative computational approach that explicitly acknowledges biological caveats while enabling the selection of high-confidence candidates for downstream experimental testing.

The current standard of care for glioblastoma involves maximal safe surgical resection followed by radiotherapy in combination with adjuvant temozolomide [2]. Although temozolomide (TMZ) is generally well tolerated, its therapeutic efficacy is limited by the rapid development of resistance and its inability to eradicate all tumor cells. These residual glioma cells ultimately lead to disease recurrence and poor prognosis [10].

In recent years, novel therapeutic strategies combining immunotherapy and viral therapies have been explored to improve antitumor efficacy while limiting systemic toxicity. Immuno-oncology approaches are designed to activate or sustain host immune responses against tumor cells; however, in glioblastoma, clinical attempts to elicit effective and durable immunity have thus far shown limited success [11]. This failure likely reflects the profound immunosuppressive microenvironment of the disease and the complex mechanisms by which glioblastoma evades immune detection. Nonetheless, defining these barriers remains essential for the rational design of next-generation strategies. Cancer vaccines, for instance, aim to induce specific immune recognition by presenting tumor-derived peptides to the immune system, even if robust clinical benefit has not yet been achieved [12].

GBM incidence increases with age and is more common in men than in women. Epidemiologically, it shows higher prevalence among individuals of Caucasian descent [4,13]. A key element in immune surveillance is the major histocompatibility complex (MHC), which presents peptides on the surface of antigen-presenting cells for T-cell recognition [14]. MHC class I molecules, in particular, present intracellular peptides that are generated via proteasomal degradation and subsequently transported into the endoplasmic reticulum by components such as endoplasmic reticulum aminopeptidase 1 (ERAP1) and transporter associated with antigen processing 1 (TAP1) [15,16]. Cytotoxic CD8+ T lymphocytes specifically recognize peptides presented in complex with MHC class I molecules on the surface of target cells [17].

Computational prediction of MHC class I-bound peptides represents a key component—among multiple complementary methods—for investigating tumor-specific T-cell immunity and prioritizing neoantigens for experimental validation. MHCflurry 2.0 is an advanced algorithm that integrates models of MHC class I binding and antigen processing to predict peptide presentation [18,19]. Its binding affinity (BA) predictor has demonstrated improved accuracy through the incorporation of mass spectrometry-derived MHC-I ligandomes [20].

Equally important for validating peptide–HLA interactions is the structural characterization of MHC–peptide complexes. Accurate prediction of quaternary protein structures is crucial for understanding protein–protein interactions. Recent advances in deep learning, particularly the development of AlphaFold2-Multimer, have significantly enhanced our ability to model such complexes with high accuracy [21,22]. AlphaFold2 has also proven capable of modeling protein–peptide interactions with remarkable precision [23]. ColabFold, a user-friendly implementation of AlphaFold2-Multimer, leverages multiple sequence alignment through MMseqs2 [24] and provides per-residue confidence scores via the predicted Local Distance Difference Test (pLDDT), offering insight into the reliability of predicted structures [25]. These structural parameters are particularly valuable when assessing the conformational integrity of peptide–MHC class I complexes.

In this study, we predicted 9-mer neoepitopes derived from recurrent somatic missense single-nucleotide variants (SNVs) found in GBM. Using MHCflurry 2.0, we evaluated the binding affinities of these mutated peptides across 13 HLA class I alleles, including 12 MHC-I supertypes and HLA-A68:01, which is of particular relevance in certain ethnic populations. Strong-binding peptides associated with recurrent GBM mutations were selected for structural validation using ColabFold to assess the folding of the peptide–HLA complex. This integrative pipeline (MHCflurry2–ColabFold) enabled the identification of structurally and immunogenically relevant neoantigens, particularly for HLA-A68:01 (including variants in *EGFR*, *TP53*, and *PIK3R1*) and HLA-B*15:01 (variant in *EGFR*), supporting their potential for personalized immunotherapeutic strategies.

## 2. Materials and Methods

### 2.1. TCGA Glioblastoma Cohort and Clinical Stratification

The GBM cohort from The Cancer Genome Atlas (TCGA) Firehose dataset was utilized for downstream analyses, including neoantigen detection, MHC class I binding affinity prediction, and three-dimensional structural modeling of peptide–HLA complexes. Clinical and genomic data for GBM-TCGA patients were retrieved from the cBioPortal for Cancer Genomics [26]. A total of 290 patients with available genomic sequencing data were included in this study (183 males, 104 females, and 3 individuals with missing gender information; see Table 1). While most anatomical and clinical variables—such as tumor size (shortest and longest dimensions), Karnofsky performance score, age at diagnosis, disease-free survival, and overall survival—did not display statistically significant sex-based differences (*p* > 0.05), genomic comparisons revealed notable disparities. Specifically, female patients exhibited a significantly higher overall mutation burden compared to males (mean mutation count: 61.4 vs. 52.7; *p* = 0.0003) and a higher nonsynonymous tumor mutational burden (TMB) (mean: 2.0 vs. 1.8; *p* = 0.00037), indicating a potential increase in genomic instability. Additionally, a modest but statistically significant difference in the fraction of the genome altered was observed between sexes (*p* = 0.0289), although the mean value remained comparable at 0.2 in both groups.

### 2.2. Identification of Missense Neoantigens and MHC Class I Binding Affinity Prediction

Single-nucleotide variants (SNVs) from the GBM-TCGA cohort were downloaded using the Table Browser tool from the University of California, Santa Cruz (UCSC) Genome Browser (annotation aligned to the human reference genome hg38) [27]. To enrich for somatic and potentially immunogenic variants, polymorphisms annotated in dbSNP were excluded, and only missense variants present in at least two patients were retained. The filtered variants were formatted into Variant Calling Format (VCF) files and functionally annotated using SnpEff version 5.2 [28], employing the MANE v1.2 transcript database for high-confidence annotation [29,30]. Only somatic variants also confirmed in the COSMIC database (version 102) were selected for downstream analysis [30]. For neoantigen prediction, mutated protein sequences were segmented using a sliding window approach to generate overlapping 9-mer peptides centered on each missense mutation. These peptides were then evaluated for their binding affinity to MHC class I molecules using MHCflurry version 2.1.5 [19]. Binding predictions were performed across 13 HLA class I alleles, including 12 broadly defined supertypes and two population-enriched alleles: HLA-A01:01, A02:01, A03:01, A24:02, A26:01, A68:01 (supertype A3, enriched in Indigenous American, South Asian, and Sub-Saharan African populations), and HLA-B07:02, B08:01, B27:05, B39:01, B40:01, B58:01, and B*15:01 (prevalent in East Asian, Oceanian, and Latin American populations) [31,32]. Predicted peptides were classified based on their binding affinity (BA) values into three categories: non-binders (BA > 500 nM), weak binders (50 nM < BA ≤ 500 nM), and strong binders (BA ≤ 50 nM).

A summary of all public databases used in this study, including their version, type of data retrieved, and reference links, is provided in Appendix A.

### 2.3. Structural Prediction of Peptide–HLA Class I Complexes

Protein sequences corresponding to the HLA-*A*68:01:01 and HLA-B15:01:01 alleles were obtained from the IPD-IMGT/HLA database (https://www.ebi.ac.uk/ipd/imgt/hla/, accessed on 18 June 2025) [33]. Structural modeling of peptide–MHC class I complexes was carried out using ColabFold, an implementation of AlphaFold2-Multimer (version 3) [34,35]. Input sequences consisted of selected strong-binding 9-mer peptides (derived from high-frequency GBM mutations) and the full-length HLA class I alpha chains, formatted as heterodimeric complexes. Sequence alignments were generated using MMseqs2, which optimizes coverage of homologous sequences to improve the quality of the multiple sequence alignment and, consequently, model accuracy [24].

The structural prediction pipeline was run using the AlphaFold2_multimer_v3_model_X, with default settings unless specified otherwise. Several confidence metrics were retrieved for model evaluation, including:−pLDDT: A per-residue score that estimates the local distance difference test (lDDT-Cα) for each amino acid [36],−Predicted TM score (pTM): Assesses global folding confidence [21],−Interface predicted TM score (ipTM): Evaluates the quality and spatial positioning of interacting chains within the complex.

Each peptide–HLA complex was modeled five times (rank_1 through rank_5), and all models were retained for structural analysis. Top-ranking models were further examined to assess the stability of the peptide–MHC interface and the structural impact of somatic mutations on binding and presentation.

### 2.4. Statistical Analysis of Quantitative Variables

Statistical analyses were conducted using R software (version 4.4.2), employing the Publish package (version 2023.01.17) for data visualization and summary reporting. Quantitative clinical and genomic variables were described using means and standard deviations. Group comparisons were performed using either parametric (Student’s *t*-test) or non-parametric (Wilcoxon rank-sum test) methods, depending on the distribution of the data. Statistical significance was defined as *p* < 0.05.

## 3. Results

### 3.1. Identification of Immunogenic Somatic Variants in Glioblastoma Through Integrated Genomic Filtering and Neoepitope Prediction

Somatic mutation data from TCGA-GBM samples were retrieved via the UCSC Genome Browser (GRCh38/hg38 assembly) and subjected to a multi-step filtering pipeline to identify potential immunogenic variants. Only missense single-nucleotide variants (SNVs) absent from dbSNP and detected in at least two patients were retained. The resulting variant list was re-annotated using SnpEff v5.2 in conjunction with the MANE 1.2 transcript database (based on the Ensembl GRCh38 assembly), allowing for standardized and high-confidence functional interpretation. To confirm the somatic origin of these mutations, additional filtering was applied against the COSMIC v102 database [30] (Table 2).

As illustrated in Figure 1A, the curated set of variants is enriched in well-established glioblastoma driver genes, including *EGFR*, *TP53*, and *PIK3R1*, consistent with their central roles in gliomagenesis. Next, overlapping 9-mer peptides encompassing each missense mutation were generated and analyzed for their binding affinity to 12 representative HLA-A and HLA-B supertypes using the MHCflurry 2.1.5 prediction algorithm.

Predicted peptide–MHC binding affinities were stratified into three categories based on nanomolar thresholds: non-binders (blue; BA > 500 nM), weak binders (green; 50 nM < BA ≤ 500 nM), and strong binders (red; BA ≤ 50 nM), as shown in Figure 1B. The identification of high-affinity binders derived from frequent somatic mutations highlights the immunotherapeutic potential of specific GBM neoepitopes, paving the way for personalized MHC class I-targeted vaccine design and immune monitoring strategies.

### 3.2. HLA-A*68:01-Mediated Neoantigen Presentation and Its Immunogenic Potential in Glioblastoma

Comprehensive analysis of predicted peptide binding across twelve representative HLA class I supertypes revealed a predominant fraction of non-binding peptides, with a smaller subset exhibiting weak or strong binding affinities (Figure 1B). The overall binding distribution (Figure 2A) indicated that supertypes such as HLA-A68:01 and HLA-B15:01 generated higher counts of strong binders, whereas alleles including HLA-A01:01, HLA-A02:01, and HLA-B*40:01 were largely dominated by non-binders. Notably, binding profiles varied considerably between HLA alleles, suggesting allele-specific differences in the capacity to present glioblastoma-associated neoepitopes. Sequence motif analysis (Figure 2B,C) further highlighted these differences: panel B illustrates alleles such as HLA-B15:01 and HLA-A68:01, which are enriched in strong binders and display conserved anchor residue motifs at position 2 and the C-terminus; in contrast, panel C shows alleles such as HLA-B27:05 and HLA-B08:01, which exhibit a more heterogeneous binding landscape with both strong and weak binders represented. These findings emphasize the heterogeneity of neoantigen presentation in glioblastoma and suggest that HLA allele-specific binding preferences may influence T-cell recognition and, ultimately, the efficacy of immunotherapeutic interventions.

To further explore the immunogenic potential of missense-derived peptides, we assessed their predicted binding affinities to two additional HLA class I alleles: HLA-A*30:01 and HLA-B*44:02 (Appendix A). This additional analysis identified two peptides as strong binders for HLA-A*30:01 (Appendix A). No strong binders were identified for the HLA-B*44:02 allele (Appendix AB). Two mutated peptides were identified as strong binders to HLA-A*30:01. These peptides originate from distinct sequence variants: one in ATRX (chrX:77618846 C>T) and the other in H3-3A (chr1:226064454 G>A), with predicted low affinities and a population frequency of 0.0051. According to their respective low frequencies in the GBM cohort, the immunogenicity of these two peptides could drive the low-influence immune response of patients.

### 3.3. Prioritization of Strong-Binding Peptides Based on HLA Allele Affinity Profiles

Several predicted neoantigenic peptides demonstrated high binding affinity to specific HLA alleles, suggesting their potential for effective immune presentation. In particular, peptides derived from *PIK3R1* (p.Gly376Arg), *EGFR* (p.Ala289Asp and p.Arg108Lys), and *TP53* (p.His179Arg and p.Thr155Asn) exhibited strong predicted affinity for HLA-A68:01 (Table 3). Conversely, the *EGFR* p.Arg108Lys peptide displayed low predicted binding to HLA-B15:01, indicating allele-specific differences in presentation potential. Peptides originating from *PDGFRA* showed diverse binding patterns across multiple HLA alleles, reflecting a degree of binding promiscuity that could increase their immunotherapeutic relevance.

These findings provide a rationale for prioritizing selected strong-binding peptides as candidates for downstream immunogenicity validation and potential incorporation into glioblastoma-targeted immunotherapy strategies.

### 3.4. Structural Modeling of the EGFR p.Ala289Asp-Derived Peptide in Complex with HLA-A*68:01

To investigate the structural basis of neoantigen presentation, we modeled the heterodimer formed between the EGFR p.Ala289Asp-derived 9-mer peptide (YSFGDTCVK) and HLA-A*68:01 using ColabFold, an accessible implementation of AlphaFold2. The analysis employed the multimer model v3, yielding a high-confidence overall structure (mean pLDDT = 79.9) and a strong interface prediction score (ipTM = 0.893), indicative of reliable peptide–MHC binding geometry.

Predicted binding affinity analysis confirmed a robust interaction, consistent with the low nanomolar binding scores observed for this peptide–allele pair (Figure 3A), supporting its high immunogenic potential. Sequence homology searches performed with the mmSeq2s database identified conserved motifs across related proteins, reinforcing the biological plausibility of this epitope (Figure 3B).

Model confidence metrics (IDDT scores) for the top five predicted structures demonstrated consistent reliability across key anchor and contact residues (Figure 3C). The contact map of the best-ranked model revealed stabilizing intramolecular interactions at the peptide–HLA interface, particularly at conserved anchor positions (Figure 3D). Visualization in both 2D and 3D formats (Figure 3E,F), including chain-colored and pLDDT-colored renderings, confirmed a coherent domain organization and high-confidence structural regions at the binding interface.

Collectively, these results validate the use of ColabFold for accurate modeling of MHC class I peptide complexes and highlight the EGFR p.Ala289Asp-derived epitope as a promising candidate for HLA-A*68:01-restricted immune targeting in glioblastoma.

### 3.5. Structural Characterization of the PIK3R1 p.Gly376Arg Mutant Peptide–HLA-A*68:01 Complex

The Gly376Arg mutation within the PIK3R1 protein presents notable immunogenic and structural implications in the context of antigen presentation by HLA-A68:01. The corresponding 9-mer mutant peptide (YTLTLRKGR) demonstrated a strong predicted binding affinity to HLA-A68:01 (48.16 nM) (Table 3, Figure 4A). Sequence alignment analysis revealed a high degree of regional conservation across PIK3R1, underscoring the potential functional importance of the mutation site (Figure 4B). Residue-level structural confidence was consistently high across the top five ranked models, as indicated by IDDT scores (Figure 4C). The contact map of the top-ranked model displayed a dense network of intra-chain interactions surrounding the mutation, suggesting localized structural perturbations that may influence peptide–MHC stability (Figure 4D). Three-dimensional structural models, visualized with chain identity and per-residue pLDDT color mapping, revealed an overall stable fold with localized flexibility at specific positions (Figure 4E). Model quality metrics (pLDDT = 80.1; ipTM = 0.867) supported both the reliability of the structural predictions and the accuracy of the peptide–HLA interface (Figure 4F). Collectively, these findings support the potential presentation of PIK3R1-derived neoantigens by HLA-A*68:01, which may be particularly relevant in individuals carrying this allele.

### 3.6. Structural Characterization of the TP53 p.His179Arg Mutant Peptide–HLA-A*68:01 Complex

The TP53 p.His179Arg mutation demonstrates notable immunological relevance by generating a 9-mer peptide (EVVRRCPHR) with high predicted binding affinity for HLA-A68:01 (34.72 nM) (Table 3, Figure 5A). This strong interaction suggests the potential formation of a stable neoantigen–HLA complex capable of eliciting T-cell-mediated immune responses. Sequence alignment analyses revealed conservation surrounding the mutation site (Figure 5B), supporting its functional significance. High per-residue IDDT scores across the top five ranked structural models (Figure 5C) confirm consistent local structural reliability. The contact map for the rank 2 model (Figure 5D) shows a well-organized network of intramolecular interactions concentrated near the mutated residue, indicating structural stabilization. Two-dimensional and three-dimensional renderings, colored by chain identity and pLDDT score (Figure 5E,F), further illustrate a coherent peptide–HLA fold with high-confidence regions at the binding interface. Model quality metrics (pLDDT: 79.9; ipTM: 0.909) indicate strong confidence in the heterodimer’s structural integrity, reinforcing the potential of the TP53 p.His179Arg variant as a functional neoepitope presented by HLA-A68:01 in individuals carrying this allele.

### 3.7. Structural Characterization of the EGFR p.Arg108Lys Mutant Peptide–HLA-B*15:01 Complex

The EGFR p.Arg108Lys mutation demonstrates immunological relevance by generating a mutant peptide (LQIIKGNMY) with a strong predicted binding affinity to HLA-B*15:01 (45.43 nM) (Table 3; Figure 6A). This suggests the potential formation of a stable neoantigen–HLA complex capable of eliciting T-cell-mediated immune responses. Sequence alignment around the mutation site reveals conserved regions (Figure 6B), supporting the structural and functional stability of the peptide. High per-residue IDDT scores across the top-ranked models (Figure 6C) further confirm the reliability of the predicted local structure. The contact map derived from the second-ranked model (Figure 6D) highlights organized intramolecular interactions, particularly in the vicinity of the mutated residue. Both 2D and 3D structural renderings, colored by chain identity and pLDDT scores (Figure 6E,F), depict a well-folded and stable peptide–HLA complex. Model performance metrics (mean pLDDT score: 80.2; ipTM: 0.926) indicate high confidence in the heterodimer interface, reinforcing the immunogenic potential of the EGFR p.Arg108Lys variant when presented by HLA-B*15:01 in individuals expressing this allele.

## 4. Discussion

In this study, we employed integrated genomic filtering and deep learning-based structural modeling to identify immunogenic somatic variants in glioblastoma multiforme (GBM) capable of forming stable peptide–HLA class I complexes. We focused specifically on alleles HLA-A68:01 and HLA-B15:01, due to their distinct binding characteristics and population distributions.

HLA-A68:01, a member of the A3 supertype, demonstrates a preference for peptides with small or aliphatic residues (Ala, Leu, Ile, Val, Met, Ser, or Thr) at position 2 and positively charged residues (Arg or Lys) at the C-terminus [37]. Our peptide motif analysis (Figure 2B, SeqLogo) confirms that candidate binders identified in this study align well with this motif, supporting the compatibility of glioblastoma-derived neoantigens with HLA-A68:01 [38]. This allele is notably enriched in Indigenous American, South Asian, and Sub-Saharan African populations [32], highlighting its potential clinical relevance across diverse genetic backgrounds.

To refine variant selection, we applied stringent filtering using databases such as dbSNP, COSMIC, and MANE, thereby enriching for high-confidence, protein-altering somatic mutations. The recurrence of mutations in canonical GBM driver genes—EGFR, TP53, and PIK3R1—supports their biological relevance and validates the pipeline’s specificity [5].

One hallmark of cancer immune evasion is impaired antigen presentation. In glioblastoma, MHC-I downregulation in GBM has been mechanistically linked to TP53 loss-of-function mutations, which transcriptionally repress key components of the antigen-processing machinery, including TAP1 and ERAP1. This reduction in antigen presentation efficiency contributes to impaired CD8^+^ T-cell recognition and immune escape [34]. However, data from an independent GBM cohort indicated that MHC-I repression was more associated with loss of TP53 expression, and neither p53 expression nor MHC-I status significantly influenced progression in Grade 4 astrocytoma [39].

Binding affinity predictions revealed substantial allele-dependent differences in neoepitope presentation. HLA-A68:01 consistently ranked among the alleles with the highest proportion of strong binders, suggesting that individuals carrying this allele may have enhanced immunological recognition of specific GBM neoantigens. Motif enrichment and anchor residue analysis further confirmed the structural compatibility between the HLA-A68:01 binding groove and key somatic peptides.

Although few clinical associations exist for HLA-A68:01, its high prevalence in Native American and African populations has attracted attention for its immunological potential [32,40]. Notably, glioblastoma incidence is 1.99 times higher among Caucasians compared to African Americans in the U.S. population [13,41,42]. In the context of infectious disease, HLA-A68:01 has been linked to protection against cardiomyopathy in Chagas disease among the Mexican population [43]. In the U.S., glioblastoma incidence among Hispanic individuals ranges from 2.42 to 2.46 per 100,000, compared to 3.45 to 3.56 in Caucasians [13,41,42].

Similarly, HLA-B*15:01 emerged as a potential presenter of the EGFR p.Arg108Lys neoepitope. This allele is enriched in East Asian, Oceanian, and Latin American populations, with particularly high frequency (29%) reported among the Ainu population in northern Japan [32,40]. In the U.S., the incidence of glioblastoma among Asians ranges from 1.18 to 1.67, substantially lower than that of Caucasians [13,41,42]. A comparative genomic study of primary GBM in Japan and Switzerland showed molecular similarities, including EGFR mutation rates of 3% (2/69) in Japan and 5% (4/81) in Switzerland. However, significant differences in EGFR codon 787 polymorphisms were observed, with G allele frequencies of 0.878 in Japan versus 0.414 in Switzerland (*p* < 0.0001) [44,45].

To complement binding affinity predictions, we applied structural modeling using AlphaFold2-multimer (ColabFold v3) to assess the conformational stability and interface integrity of peptide–HLA complexes. This deep learning-based approach confirmed efficient incorporation of 9-mer neoantigen peptides between the α- and β-chains of the respective MHC-I alleles. Structural characterization of high-affinity peptides derived from EGFR, TP53, and PIK3R1 revealed coherent folding patterns, stable local interactions, and high-confidence metrics (e.g., pLDDT and ipTM scores). These findings highlight the potential of rapid structure prediction tools to aid in the prioritization of clinically actionable neoantigens.

Overall, this study underscores the utility of integrating genomic, immunological, and structural data to predict immunogenic neoantigens in glioblastoma. Our results suggest that population-specific HLA alleles such as HLA-A68:01 and HLA-B15:01 may influence neoantigen presentation landscapes and inform future immunotherapeutic strategies tailored to diverse genetic backgrounds.

### Limitations and Future Directions

Several limitations should be acknowledged. First, this study is based exclusively on computational analyses and therefore provides in silico predictions that require empirical validation before translational application. Although the integrative use of MHCflurry 2.0 and AlphaFold2-Multimer enables robust prioritization of candidate neoantigens, these approaches do not confirm actual peptide processing, HLA presentation, or T-cell recognition in biological systems. Experimental assays such as peptide–HLA refolding and stability measurements, surface plasmon resonance or microscale thermophoresis to assess binding kinetics, mass-spectrometry-based immunopeptidomics, and functional T-cell activation assays (ELISpot, intracellular cytokine staining) will be essential to validate the computationally identified candidates.

Second, the predictive models employed here—while integrating antigen-processing features—cannot yet fully recapitulate proteasomal cleavage specificity, TAP transport efficiency, post-translational modifications, or the influence of the tumor microenvironment on antigen presentation. Incorporating these factors, along with patient-specific expression data, will refine future predictions.

Finally, although the HLA-I allelic panel used in this study covers the major human supertypes and includes population-enriched variants, the inclusion of additional alleles and the integration of patient-matched HLA genotypes will improve population coverage and translational relevance. These methodological extensions, together with experimental validation, represent the next step toward comprehensive neoantigen discovery in glioblastoma, in accordance with current recommendations for preclinical modeling of patient heterogeneity [9].

Although the integration of AlphaFold2-Multimer into neoantigen prediction provides valuable atomic-level insight, several intrinsic limitations of this structural modeling approach must be considered. First, AlphaFold2-based predictions represent static conformational snapshots that do not capture the dynamic flexibility of peptide–MHC-I complexes, the kinetics of peptide loading and release, or conformational adjustments induced by T-cell receptor (TCR) engagement [22]. This oversimplification may influence estimates of complex stability and TCR accessibility.

Second, the current models exclude key biological cofactors such as β_2_-microglobulin, chaperones involved in peptide loading (tapasin, calreticulin), and the lipid-membrane context that shapes the orientation of the HLA groove. These omissions can subtly alter inter-chain geometry and solvent exposure of peptide side chains relevant for TCR contact [21]. Moreover, post-translational modifications—including phosphorylation, oxidation, or cysteinylation—are not represented, although they can critically modulate peptide processing and immunogenicity [19].

Third, the confidence metrics reported by AlphaFold2-Multimer (pLDDT, ipTM, and PAE scores) are computational proxies for structural accuracy rather than direct indicators of biophysical stability or immunological relevance. High-confidence predictions do not necessarily imply that a given peptide–MHC will form in vivo or elicit a T-cell response. Therefore, the structural models presented here should be interpreted as hypothesis-generating tools that assist in candidate prioritization rather than as definitive representations of native complexes.

Experimental validation remains essential to confirm these predictions. Future work will employ biophysical assays such as peptide–HLA refolding and thermostability (Tm) measurements, surface-plasmon-resonance or microscale-thermophoresis binding kinetics analyses, and mass-spectrometry-based identification of naturally presented peptides to verify complex formation and stability. Integrating these empirical data with computational refinement will improve the predictive reliability of structural modeling in neoantigen discovery.

The tumor and immune microenvironment (TME) play decisive roles in determining the efficacy of immunotherapies in glioblastoma. GBM is characterized by extensive infiltration of tumor-associated macrophages and microglia that frequently acquire an M2-like, immunosuppressive phenotype; these cells secrete cytokines such as IL-10 and TGF-β, inhibit cytotoxic T-cell activity, and promote tumor progression [46]. In parallel, the TME exhibits profound T-cell exhaustion, limited trafficking of effector lymphocytes, and expression of immune-checkpoint ligands such as PD-L1 and TIM-3 [47,48]. These features substantially constrain the immune visibility of neoantigens, even when strong peptide–MHC-I binding is predicted computationally. Although our present study focuses on the structural and binding aspects of neoepitope–HLA interactions, future work should incorporate TME variables derived from transcriptomic deconvolution or single-cell RNA-seq datasets to model immune infiltration, macrophage polarization states, and cytokine signaling networks (e.g., through CIBERSORTx or TIMER analyses). Integration of these multi-omic layers with the current structural pipeline will enable prioritization of neoantigens that are not only strong HLA binders but also likely to be presented in an immune-permissive microenvironment.

## 5. Conclusions

Through integrative genomic filtering, neoepitope prediction, and structure-based modeling, this study identifies glioblastoma-derived somatic mutations with high immunogenic potential for HLA class I presentation—particularly via HLA-A68:01 and HLA-B15:01. Our findings highlight key driver mutations in EGFR, TP53, and PIK3R1 as promising sources of strong-binding neoantigens and demonstrate allele-specific variability in binding affinity across the HLA landscape.

By employing AlphaFold2-multimer via ColabFold, we provide robust structural evidence supporting the stability and feasibility of peptide–MHC class I complexes. These insights reinforce the therapeutic relevance of these neoepitopes, especially in tumors with intact MHC-I expression.

Overall, this approach establishes a foundation for rational neoantigen prioritization and paves the way for the development of personalized immunotherapies tailored to individual HLA genotypes and tumor mutational profiles.

## Figures and Tables

**Figure 1 biomedicines-13-02715-f001:**
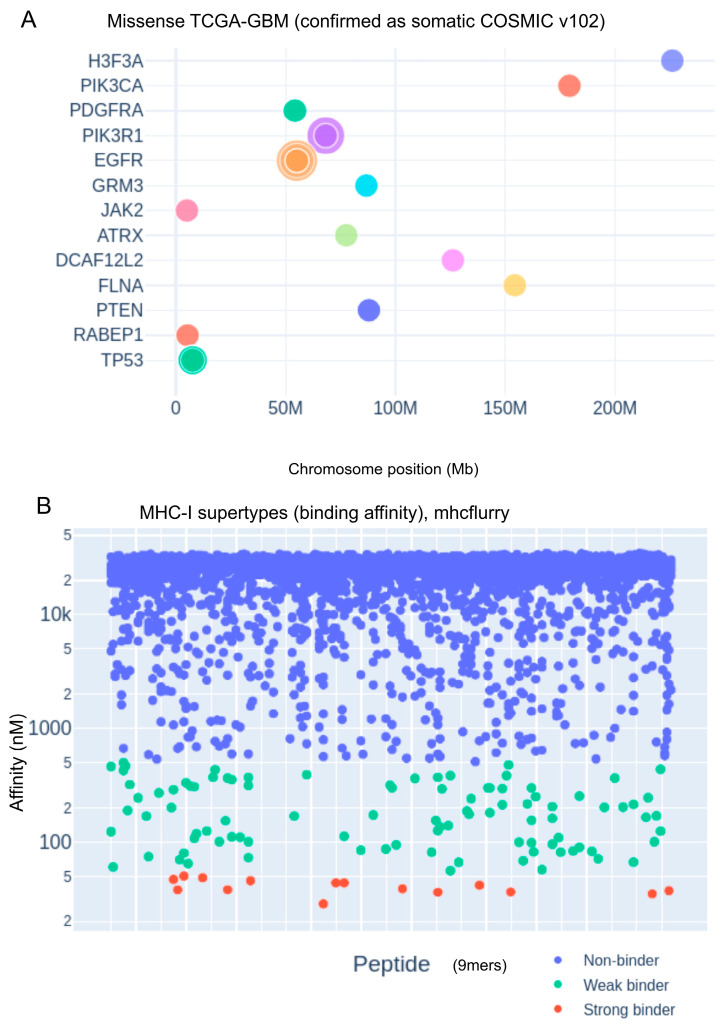
Genomic distribution of glioblastoma-associated mutations and predicted MHC class I peptide binding affinities. (**A**) Genomic landscape of missense mutations in TCGA-GBM samples, validated as somatic in the COSMIC database (version 102). Mutations are plotted according to chromosomal location (in Mb), with key glioblastoma-associated genes such as EGFR, TP53, and PTEN highlighted. (**B**) Predicted binding affinities of 9-mer peptides to representative MHC class I supertypes, calculated using MHCflurry. Peptides are classified by binding strength based on nanomolar affinity thresholds: non-binders (blue), weak binders (green), and strong binders (red).

**Figure 2 biomedicines-13-02715-f002:**
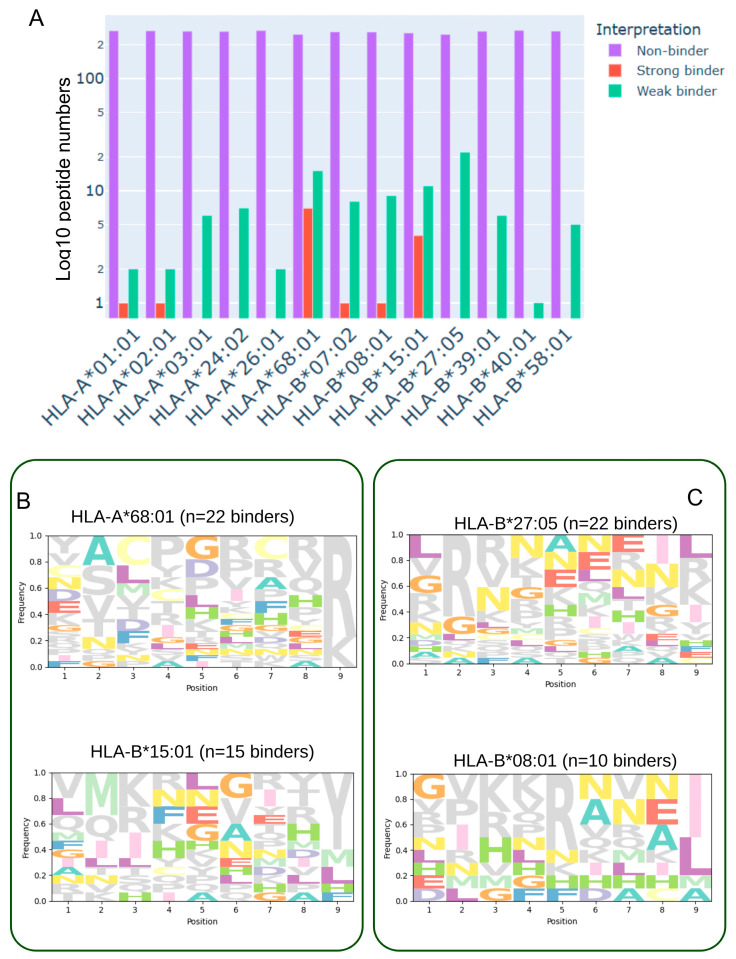
Binding affinity distribution and sequence motifs of predicted glioblastoma-associated neoantigens across HLA class I supertypes. (**A**) Histogram showing the distribution of 9-mer peptides, derived from single-nucleotide variants (SNVs) in the TCGA-GBM cohort, across twelve representative HLA class I supertypes and HLA-A68:01. Peptides were generated using a sliding window around each SNV and evaluated for binding affinity with MHCflurry2. Bars are color-coded according to predicted binding category: strong binders (red, BA < 50 nM), weak binders (green, 50 nM ≤ BA ≤ 500 nM), and non-binders (purple, BA > 500 nM). (**B**) Sequence logos depicting amino acid preferences among strong-binding peptides for selected HLA alleles (e.g., HLA-B15:01 and HLA-A68:01), showing conserved anchor residues at position 2 and the C-terminus. (**C**) Sequence logos illustrating peptide motifs for HLA alleles with mixed binding profiles (e.g., HLA-B27:05 and HLA-B*08:01), comprising both strong and weak binders and reflecting greater motif diversity.

**Figure 3 biomedicines-13-02715-f003:**
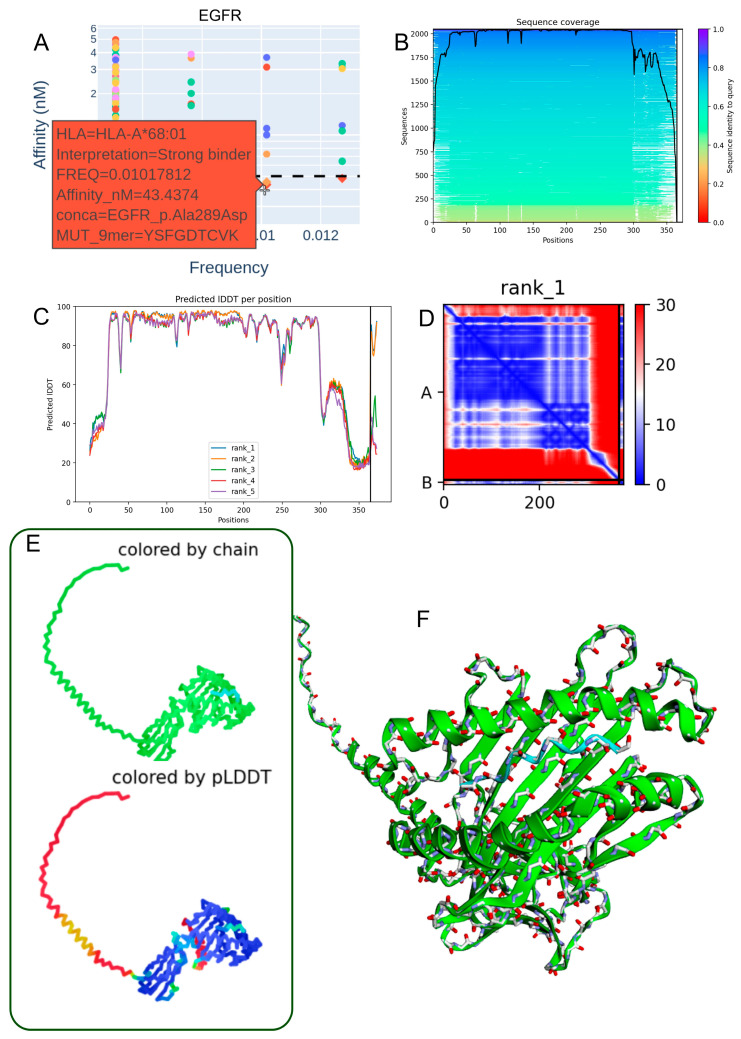
Structural modeling of the EGFR p.Ala289Asp-derived peptide in complex with HLA-A*68:01. (**A**) Predicted binding affinity (in the low nanomolar range) indicating strong interaction between the EGFR p.Ala289Asp peptide (YSFGDTCVK) and the HLA-A*68:01 allele. (**B**) Sequence alignment from the mmSeq2s database showing conserved regions across homologous proteins. (**C**) Per-residue IDDT confidence scores for the top five ranked structural models. (**D**) Contact map of the top-ranked model highlighting intramolecular interactions at the peptide–HLA interface. (**E**) Two-dimensional structural representation colored by chain or pLDDT score, with model reliability ranging from red (low confidence) to blue (high confidence). (**F**) Three-dimensional visualization of the heterodimer colored by chain, illustrating molecular topology.

**Figure 4 biomedicines-13-02715-f004:**
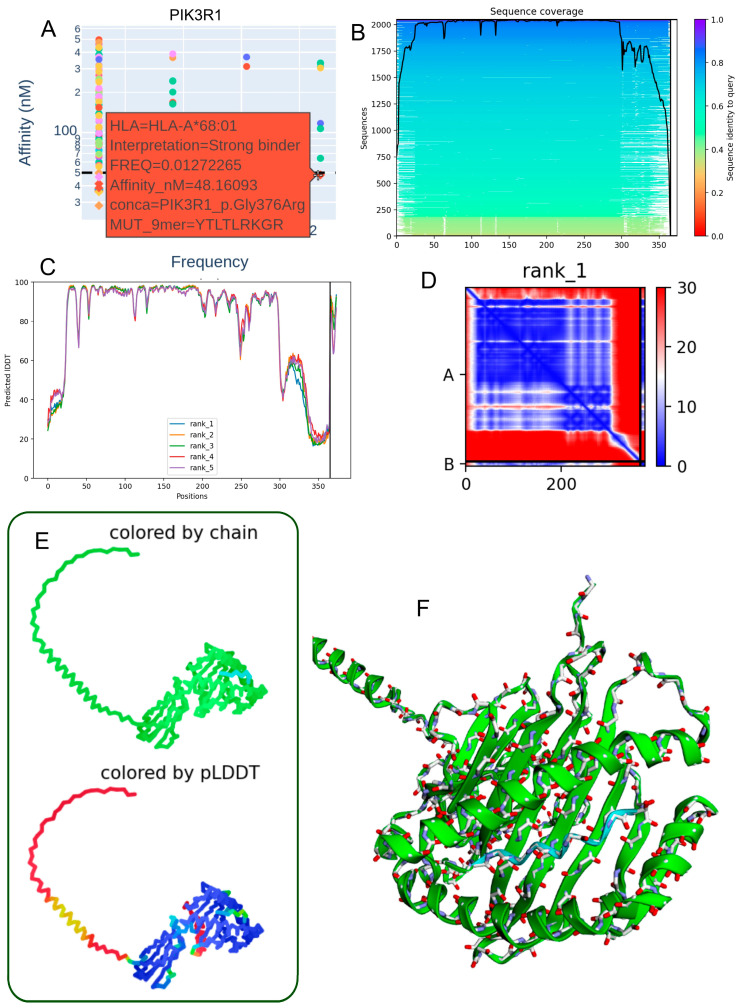
Predicted 3D structure of the heterodimer formed between the PIK3R1 p.Gly376Arg mutant peptide and the HLA-A68:01 molecule. (**A**) Binding affinity prediction (nanomolar range) indicating strong interaction of the PIK3R1 p.Gly376Arg-derived peptide (YTLTLRKGR) with the HLA-A68:01 allele. (**B**) Sequence alignment showing conserved regions across homologous sequences in the mmSeq2s database. (**C**) Per-residue IDDT confidence scores for the top five ranked structural models. (**D**) Contact map from the top-ranked model depicting intramolecular interactions at the peptide–HLA interface. (**E**) Two-dimensional structural representation colored by chain or pLDDT score, with reliability ranging from red (low confidence) to blue (high confidence). (**F**) Three-dimensional visualization of the heterodimer colored by chain, illustrating molecular topology.

**Figure 5 biomedicines-13-02715-f005:**
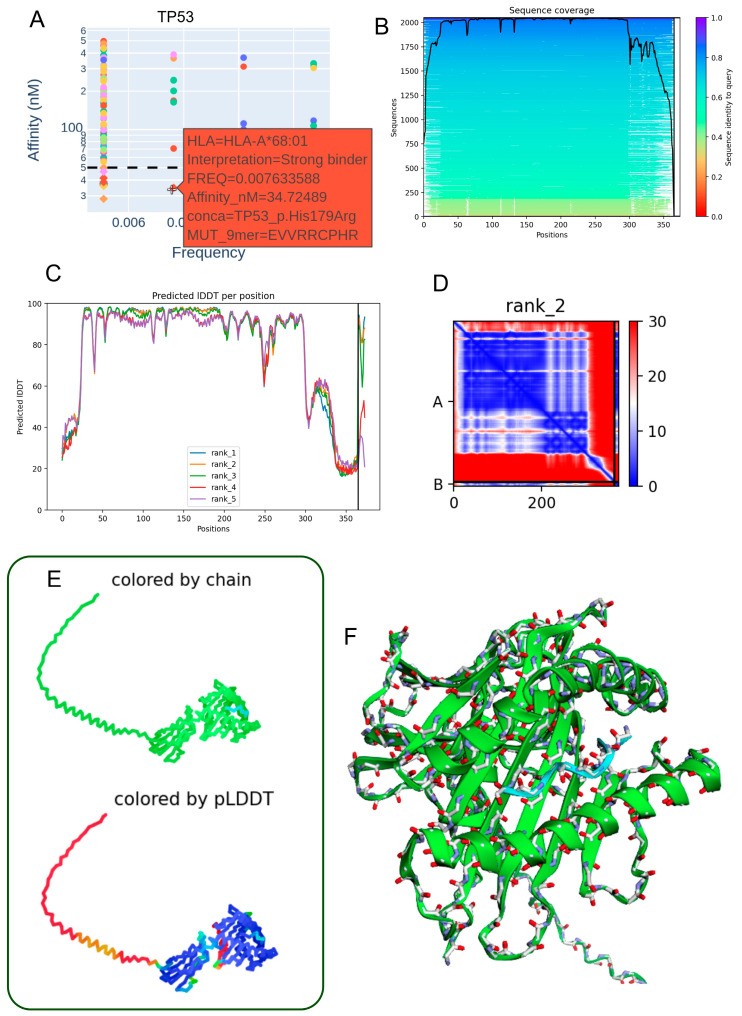
Predicted three-dimensional structure of the heterodimer formed between the TP53 p.His179Arg mutant-derived peptide and the HLA-A68:01 molecule. (**A**) The TP53 p.His179Arg peptide (EVVRRCPHR) exhibits a low allelic frequency and a strong predicted binding affinity (34.72 nM) to the HLA-A68:01 allele, suggesting a stable interaction. (**B**) Sequence alignment highlights conserved regions between the mutant peptide and homologous sequences retrieved from the mmSeq2s database. (**C**) Per-residue IDDT scores for the top five ranked structural models indicate high prediction reliability. (**D**) Contact map derived from the second-ranked model reveals key intramolecular residue interactions proximal to the mutation site. (**E**) Two-dimensional structural representation colored by chain or pLDDT score (ranging from red: low confidence to blue: high confidence), illustrating model robustness. (**F**) Three-dimensional visualization of the peptide–HLA complex, with chain-specific coloring depicting the molecular topology.

**Figure 6 biomedicines-13-02715-f006:**
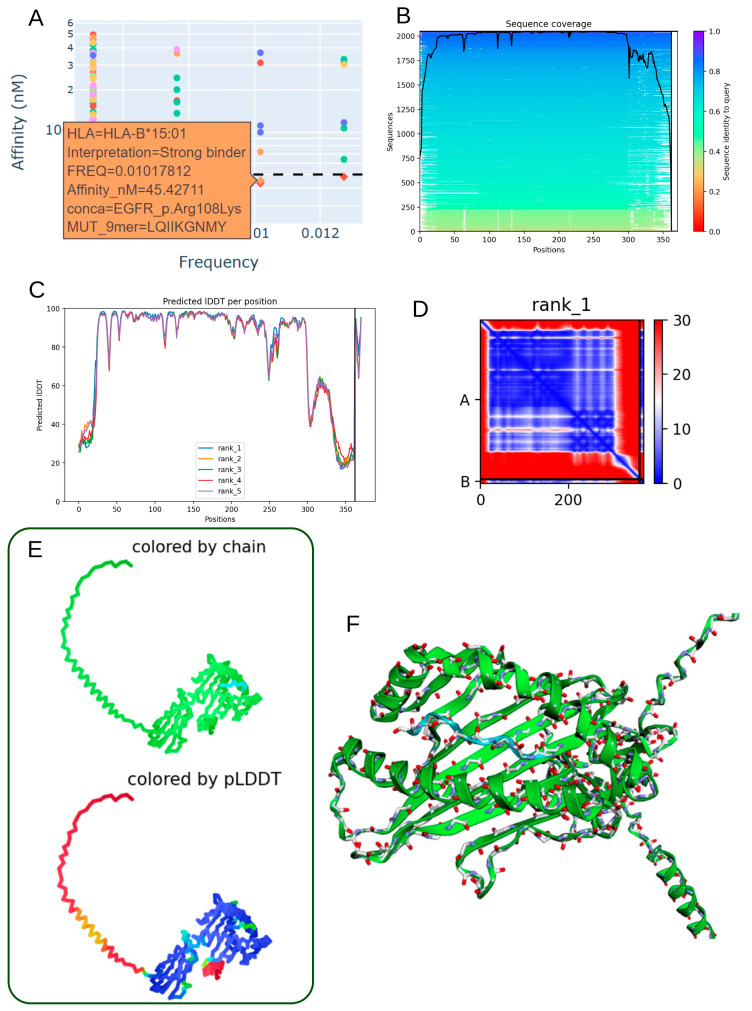
Predicted 3D structure of the heterodimer formed between the EGFR p.Arg108Lys mutant-derived peptide and HLA-B*15:01 molecule. (**A**) Low allelic frequency and strong-binding affinity prediction (45.43 nM) indicate robust interaction of the EGFR p.Arg108Lys peptide (LQIIKGNMY) with the HLA-B*15:01 allele. (**B**) Sequence alignment reveals conserved regions between the mutant peptide and reference sequences from the mmSeq2s database. (**C**) Per-residue IDDT scores plotted for the top five ranked structural models, confirming prediction reliability. (**D**) Contact map from rank 1 model displaying key intramolecular residue interactions surrounding the mutation site. (**E**) 2D structure colored by chain or pLDDT score, from red (low confidence) to blue (high confidence), illustrating model robustness. (**F**) 3D visualization of the peptide–HLA complex showing molecular topology by chain coloration.

**Table 1 biomedicines-13-02715-t001:** TCGA-GBM multiform cohort n = 290 patients with genomic data (3 missing data for gender stratification): (tmb) tumor mutation burden.

Variable	Level	Male (*n* = 183)	Female (*n* = 104)	Total (*n* = 290)	*p*-Value
Tumor shortest_dimension	mean (sd)	0.5 (0.2)	0.5 (0.2)	0.5 (0.2)	0.8311346
Tumor longest_dimension	mean (sd)	1.1 (0.4)	1.1 (0.4)	1.1 (0.4)	0.6930486
mutation_count	mean (sd)	52.7 (18.9)	61.4 (21.1)	55.8 (20.2)	0.0003163
tmb_nonsynonymous	mean (sd)	1.8 (0.6)	2 (0.7)	1.9 (0.7)	0.0003700
fraction_genome_altered	mean (sd)	0.2 (0.1)	0.2 (0.1)	0.2 (0.1)	0.0289327
karnofsky_performance score	mean (sd)	76.2 (18)	74.6 (17.1)	75.7 (17.6)	0.5465281
disease_free_months	mean (sd)	8.9 (10.4)	9.1 (7.9)	9 (9.6)	0.8649928
diagnosis_age	mean (sd)	60.5 (12.9)	61.9 (13.2)	61 (13)	0.3792085
overall_survival_months	mean (sd)	13.6 (12.9)	15.7 (16.5)	14.3 (14.3)	0.2242429

**Table 2 biomedicines-13-02715-t002:** Missense single-nucleotide variants (SNVs) identified in at least two distinct patients from the GBM-TCGA cohort and confirmed as somatic mutations in the COSMIC database (version 102). For each variant, the following information is provided: gene symbol (Gene), COSMIC legacy identifier (Mutation ID), amino acid substitution (Mutation AA), nucleotide change (Mutation CDS), and the frequency of occurrence within the cohort (Frequency). Variants are ranked in descending order of frequency.

Gene	Mutation_ID	Mutation_AA	Mutation_CDS	Frequency
EGFR	COSM191968	p.R222C	c.664C>T	1.53 × 10^2^
PIK3R1	COSM1438284	p.G376R	c.1126G>A	1.27 × 10^2^
EGFR	COSM4970170	p.R108K	c.323G>A	1.02 × 10^2^
EGFR	COSM21685	p.A289D	c.866C>A	1.02 × 10^2^
EGFR	COSM1559807	p.R252C	c.754C>T	7.63 × 10^3^
EGFR	COSM6919374	p.T263P	c.787A>C	7.63 × 10^3^
TP53	COSM3403255	p.C275Y	c.824G>A	7.63 × 10^3^
TP53	COSM10889	p.H179R	c.536A>G	7.63 × 10^3^
H3F3A	COSM327929	p.G35R	c.103G>A	5.09 × 10^3^
PIK3CA	COSM12591	p.M1043V	c.3127A>G	5.09 × 10^3^
PDGFRA	COSM3409357	p.E229K	c.685G>A	5.09 × 10^3^
PDGFRA	COSM3409362	p.L655F	c.1965G>C	5.09 × 10^3^
PIK3R1	COSM3410369	p.K379N	c.1137A>T	5.09 × 10^3^
EGFR	COSM6951164	p.G63R	c.187G>A	5.09 × 10^3^
EGFR	COSM7484095	p.Y270C	c.809A>G	5.09 × 10^3^
EGFR	COSM2149971	p.H304Y	c.910C>T	5.09 × 10^3^
EGFR	COSM2155593	p.T363I	c.1088C>T	5.09 × 10^3^
GRM3	COSM218933	p.R183C	c.547C>T	5.09 × 10^3^
JAK2	COSM29104	p.R564Q	c.1691G>A	5.09 × 10^3^
ATRX	COSM1716715	p.R1803H	c.5408G>A	5.09 × 10^3^
DCAF12L2	COSM385381	p.P334L	c.1001C>T	5.09 × 10^3^
DCAF12L2	COSM1315186	p.R246H	c.737G>A	5.09 × 10^3^
FLNA	COSM3406147	p.V1240M	c.3718G>A	5.09 × 10^3^
PTEN	COSM5135	p.G36R	c.106G>A	5.09 × 10^3^
PTEN	COSM5045	p.S170N	c.509G>A	5.09 × 10^3^
RABEP1	COSM7296694	p.T279M	c.836C>T	5.09 × 10^3^
TP53	COSM4781979	p.D281A	c.842A>C	5.09 × 10^3^
TP53	COSM10943	p.D281H	c.841G>C	5.09 × 10^3^
TP53	COSM99626	p.C238F	c.713G>T	5.09 × 10^3^
TP53	COSM11218	p.T155N	c.464C>A	5.09 × 10^3^

**Table 3 biomedicines-13-02715-t003:** Predicted binding affinities of neoantigenic peptides across common HLA class I alleles. The table lists 16 cancer-associated somatic mutations identified in the GBM-TCGA cohort, each represented by a 9-mer mutant peptide (MUT_9mer). For each mutation, the table reports its cohort frequency, amino acid position, and COSMIC legacy ID. Predicted binding affinities (nM) to multiple HLA class I alleles were computed using MHCflurry2. Mutations are ordered by decreasing cohort frequency.

Identifier	Frequency	Mut_9mer	Position	Legacy Mutation Id	Hla	Affinity_Nm
PIK3R1_p.Gly376Arg	1.27 × 10^2^	YTLTLRKGR	9	COSM1438284	HLA-A*68:01	48.161
EGFR_p.Arg108Lys	1.02 × 10^2^	LQIIKGNMY	5	COSM4970170	HLA-B*15:01	45.427
EGFR_p.Ala289Asp	1.02 × 10^2^	DTCVKKCPR	1	COSM21685	HLA-A*68:01	43.446
EGFR_p.Ala289Asp	1.02 × 10^2^	YSFGDTCVK	5	COSM21685	HLA-A*68:01	43.437
TP53_p.His179Arg	7.63 × 10^3^	EVVRRCPHR	9	COSM10889	HLA-A*68:01	34.725
PDGFRA_p.Leu655Phe	5.09 × 10^3^	GPHFNIVNL	4	COSM3409362	HLA-B*07:02	46.540
PDGFRA_p.Leu655Phe	5.09 × 10^3^	HLGPHFNIV	6	COSM3409362	HLA-A*02:01	37.647
PDGFRA_p.Leu655Phe	5.09 × 10^3^	IMTHLGPHF	9	COSM3409362	HLA-B*15:01	49.837
EGFR_p.Gly63Arg	5.09 × 10^3^	VLRNLEITY	3	COSM6951164	HLA-B*15:01	37.790
EGFR_p.Tyr270Cys	5.09 × 10^3^	LMLCNPTTY	4	COSM7484095	HLA-B*15:01	28.328
GRM3_p.Arg183Cys	5.09 × 10^3^	LSDKSCYDY	6	COSM218933	HLA-A*01:01	38.499
ATRX_p.Arg1803His	5.09 × 10^3^	VMKKHAHIL	5	COSM1716715	HLA-B*08:01	35.925
DCAF12L2_p.Arg246His	5.09 × 10^3^	PVYAHIHPR	7	COSM1315186	HLA-A*68:01	41.433
PTEN_p.Gly36Arg	5.09 × 10^3^	IAMRFPAER	4	COSM5135	HLA-A*68:01	36.127
TP53_p.Thr155Asn	5.09 × 10^3^	DSTPPPGNR	8	COSM11218	HLA-A*68:01	36.966

## Data Availability

Python 3.13.0scripts and data generated during this study are accessible at the following web address: https://github.com/cdesterke/GBM_MHCI.

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
