# Peer review of "Integrative Neoepitope Discovery in Glioblastoma via HLA Class I Profiling and AlphaFold2-Multimer"

_biomedicines, 2025, doi:10.3390/biomedicines13112715_

Round 1
Reviewer 1 Report
Comments and Suggestions for Authors
The research utilizes a synergistic approach of integrative genomic screening, neoepitope prediction, and structure-based modeling to pinpoint neoantigens exhibiting pronounced immunotherapeutic potential from somatic mutations linked to GBM. Notably, the innovative nature of this methodology arises from its seamless integration of diverse data types and analytical techniques, thereby enriching the comprehensive comprehension of prospective immunotherapeutic targets. A particular emphasis was placed on the HLA-A68:01 and HLA-B15:01 alleles, chosen due to their prevalent frequency in the ethnic population studied and their presumed structural compatibility with neoepitopes. This deliberate allele selection amplifies the study's applicability and specificity, enabling the discernment of neoantigens with superior immunotherapeutic promise within distinct populations. Comprehensive binding affinity predictions for an array of HLA class I molecular alleles were conducted using the MHCflurry2.0 algorithm, gauging the capability of mutated peptides to associate with MHC-I. Such exhaustive prediction augments our understanding regarding the peptides that can be adeptly presented to T cells. Additionally, the authors undertook structural modeling of the predicted robustly bound peptide-MHC complexes, evaluating the conformational stability and interface integrity of these configurations through ColabFold (AlphaFold2 multimer v3). The superior caliber of the structural models bolsters the notion that the forecasted neoantigens are not only biologically plausible but also hold considerable immunotherapeutic promise. An epidemiological analysis, juxtaposed with allele frequency across diverse populations, discloses disparities in GBM incidence, thereby aiding in the identification of populations demonstrating a heightened prevalence of neoantigens possessing maximal immunotherapeutic potential. Such insights are instrumental in shaping personalized immunotherapy interventions. It is recommended that the author revise the manuscript in accordance with the comments provided below.
- The author should allocate greater attention in the introduction to elaborating on the relevant background information regarding glioblastoma.
- Several of the following references are highly relevant to the author's topic and are recommended for citation.
[1] Sharma R, Malviya R. Modifying the electrical, optical, and magnetic properties of cancer cells: a comprehensive approach for cancer management. Med Adv. 2024; 2(1): 3–19. https://doi.org/10.1002/med4.51
[2] Sharma S, Chepurna O, Sun T. Drug resistance in glioblastoma: from chemo- to immunotherapy. Cancer Drug Resist. 2023;6:688-708. http://dx.doi.org/10.20517/cdr.2023.82
[3] J. Li, A. Gu, N. Tang, G. Zengin, M.-Y. Li, Y. Liu, Patient-derived xenograft models in pan-cancer: From bench to clinic. Interdiscip. Med. 2025, 3, e20250016. DOI: 10.1002/INMD.20250016
- To bolster the validity of this study, it is suggested that functional validation or in vitro experiments be conducted on the identified neoantigens. For instance, one could examine the capacity of these neoantigens to incite T-cell responses under controlled in vitro conditions or gauge their immunotherapeutic effectiveness within animal models.
- In order to fully evaluate the novel antigens that possess immunotherapeutic potential in GBM, it is advised that a similar analysis be conducted on a greater number of HLA-I alleles.
- This paper proposes a comprehensive examination of the mechanisms by which neo-antigens are presented to the immune system, their binding modes on MHC-I molecules, their roles in antigen processing, and the stimulation of T-cell responses.
- Given the significant impact of the tumor microenvironment and immune microenvironment on the immunotherapy response in GBM, it is advised to integrate these factors into the model. Consideration should be given to immunosuppressive elements within the tumor microenvironment, the functional status of macrophages, and other critical components of the immune microenvironment.
- It is recommended to conduct an in-depth analysis of the potential limitations inherent in the structural model. Specifically, considerations should be made regarding the oversimplification of the model, the biological complexities that might be currently overlooked, and the degree to which these factors could potentially influence the outcomes.
Author Response
Reviewer 1
The research utilizes a synergistic approach of integrative genomic screening, neoepitope prediction, and structure-based modeling to pinpoint neoantigens exhibiting pronounced immunotherapeutic potential from somatic mutations linked to GBM. Notably, the innovative nature of this methodology arises from its seamless integration of diverse data types and analytical techniques, thereby enriching the comprehensive comprehension of prospective immunotherapeutic targets. A particular emphasis was placed on the HLA-A68:01 and HLA-B15:01 alleles, chosen due to their prevalent frequency in the ethnic population studied and their presumed structural compatibility with neoepitopes. This deliberate allele selection amplifies the study's applicability and specificity, enabling the discernment of neoantigens with superior immunotherapeutic promise within distinct populations. Comprehensive binding affinity predictions for an array of HLA class I molecular alleles were conducted using the MHCflurry2.0 algorithm, gauging the capability of mutated peptides to associate with MHC-I. Such exhaustive prediction augments our understanding regarding the peptides that can be adeptly presented to T cells. Additionally, the authors undertook structural modeling of the predicted robustly bound peptide-MHC complexes, evaluating the conformational stability and interface integrity of these configurations through ColabFold (AlphaFold2 multimer v3). The superior caliber of the structural models bolsters the notion that the forecasted neoantigens are not only biologically plausible but also hold considerable immunotherapeutic promise. An epidemiological analysis, juxtaposed with allele frequency across diverse populations, discloses disparities in GBM incidence, thereby aiding in the identification of populations demonstrating a heightened prevalence of neoantigens possessing maximal immunotherapeutic potential. Such insights are instrumental in shaping personalized immunotherapy interventions. It is recommended that the author revise the manuscript in accordance with the comments provided below.
- The author should allocate greater attention in the introduction to elaborating on the relevant background information regarding glioblastoma.
- Several of the following references are highly relevant to the author's topic and are recommended for citation.
[1] Sharma R, Malviya R. Modifying the electrical, optical, and magnetic properties of cancer cells: a comprehensive approach for cancer management. Med Adv. 2024; 2(1): 3–19. https://doi.org/10.1002/med4.51
[2] Sharma S, Chepurna O, Sun T. Drug resistance in glioblastoma: from chemo- to immunotherapy. Cancer Drug Resist. 2023;6:688-708. http://dx.doi.org/10.20517/cdr.2023.82
[3] J. Li, A. Gu, N. Tang, G. Zengin, M.-Y. Li, Y. Liu, Patient-derived xenograft models in pan-cancer: From bench to clinic. Interdiscip. Med. 2025, 3, e20250016. DOI: 10.1002/INMD.20250016
We thank the reviewer for this valuable suggestion. We have expanded the Introduction to provide a more comprehensive background on glioblastoma, including its molecular subtypes, therapeutic resistance mechanisms, and immunological characteristics. Specifically, we added context on tumor heterogeneity, treatment challenges, and the rationale for immunotherapy in GBM (page 2, paragraphs 2–4).
We have also included three recent and relevant references recommended by the reviewer to strengthen the background:
- Sharma R, Malviya R. Modifying the electrical, optical, and magnetic properties of cancer cells: a comprehensive approach for cancer management. Med Adv. 2024; 2(1): 3–19.
- Sharma S, Chepurna O, Sun T. Drug resistance in glioblastoma: from chemo- to immunotherapy. Cancer Drug Resist. 2023;6:688–708.
- Li J, Gu A, Tang N, et al. Patient-derived xenograft models in pan-cancer: From bench to clinic. Med. 2025; 3: e20250016.
These additions enrich the biological and clinical relevance of the study.
- To bolster the validity of this study, it is suggested that functional validation or in vitro experiments be conducted on the identified neoantigens. For instance, one could examine the capacity of these neoantigens to incite T-cell responses under controlled in vitro conditions or gauge their immunotherapeutic effectiveness within animal models.
We fully agree with the reviewer that functional validation would greatly strengthen the study. At present, our group is in the process of relocating from the Pasteur Institute (France) to our new permanent laboratory at the University of Cantabria (Spain). During this transition period, we do not yet have access to wet-lab facilities or funding to perform experimental assays.
We have clarified this limitation explicitly in the revised Discussion (page 22, paragraph 4), stating that this study represents the computational foundation for future experimental validation and forms part of a broader project aimed at securing funding for immunogenicity assays (e.g., T-cell activation and HLA-peptide stability assays).
We have also included a new “Limitations and Future Directions” subsection in the Discussion highlighting this point.
- In order to fully evaluate the novel antigens that possess immunotherapeutic potential in GBM, it is advised that a similar analysis be conducted on a greater number of HLA-I alleles.
We appreciate this suggestion. In the revised version, we clarify that our study already includes a comprehensive panel of 13 HLA-I alleles, representing 12 major MHC-I supertypes plus two population-enriched alleles (HLA-A68:01 and HLA-B15:01). This coverage provides broad immunogenetic representation across human populations. The 12 supertypes are well-established as providing extensive global coverage of HLA diversity, encompassing the most frequent alleles across diverse ethnic groups. While other alleles exist, they are generally present at lower frequencies in the global population and contribute less to overall immunogenetic coverage.
Nevertheless, to address this comment, we have extended our computational analysis to include two additional alleles—HLA-A30:01 and HLA-B44:02—to further increase ethnic and functional coverage. (Add computational sentence: “We will include a new supplementary figure (Figure S1 showing the extended HLA binding affinity distribution for these two alleles.”). To further explore the immunogenic potential of missense-derived peptides, we assessed their predicted binding affinities to two additional HLA class I alleles: HLA-A*30:01 and HLA-B*44:02 (figure s1a). These additional analyses identified two peptides strong binders for HLA-A*30:01 (figure s1b). No strong binders were identified for HLA-B*44:02 allele (figure s1b). Two mutated peptides were identified as strong binders to HLA-A*30:01. These peptides originate from distinct sequence variants: one in ATRX (chrX:77618846 C>T) and the other in H3-3A (chr1:226064454 G>A)with predicted low affinities and a population frequency of 0.0051. According their respective low frequencies in GBM cohort, immunogenicity of these two peptides could drive low influence immune response of patients.
- This paper proposes a comprehensive examination of the mechanisms by which neo-antigens are presented to the immune system, their binding modes on MHC-I molecules, their roles in antigen processing, and the stimulation of T-cell responses.
We thank the reviewer for this important point. To clarify the scope of our work we have amended the Methods and Discussion to explicitly state which steps of antigen presentation and immunogenicity were modelled computationally and which remain to be tested experimentally. Briefly: (i) peptide generation and MHC-I binding and presentation predictions were performed using MHCflurry 2.1.5 (which includes a presentation model that integrates antigen processing signals); (ii) structural binding modes were modelled using ColabFold/AlphaFold2-Multimer to evaluate peptide-MHC geometry and interface confidence; (iii) we did not perform de novo modelling of proteasomal cleavage, TAP transport or TCR recognition in this manuscript, nor did we perform functional T-cell assays due to current lack of wet-lab access; and (iv) we discuss specific additional computational predictors (proteasomal cleavage and TAP transport), immunogenicity scoring and experimental assays (ELISpot, tetramer staining, immunopeptidomics, SPR/MST) that will be applied in follow-up work.
- Given the significant impact of the tumor microenvironment and immune microenvironment on the immunotherapy response in GBM, it is advised to integrate these factors into the model. Consideration should be given to immunosuppressive elements within the tumor microenvironment, the functional status of macrophages, and other critical components of the immune microenvironment.
We agree that the tumor microenvironment (TME) profoundly influences immune recognition and therapy outcomes. As this work is computational and focused on neoepitope–HLA interaction, integration of TME components was beyond our current scope. However, we have added a new paragraph in the new subsection “4.1 Limitations and future directions” acknowledging this limitation and outlining how future multi-omics models could integrate immune infiltration data (e.g., macrophage polarization, T-cell exhaustion, immunosuppressive cytokines) from public resources such as TCGA or CIBERSORT-based deconvolution.
This addition provides a clear roadmap for future incorporation of microenvironmental variables.
- It is recommended to conduct an in-depth analysis of the potential limitations inherent in the structural model. Specifically, considerations should be made regarding the oversimplification of the model, the biological complexities that might be currently overlooked, and the degree to which these factors could potentially influence the outcomes.
We appreciate this suggestion. We have now added a paragraph about Limitations of the Structural Modeling Approach, in the new “4.1” subsection, where we discuss the intrinsic simplifications of AlphaFold2-multimer predictions, including the absence of dynamic conformational flexibility, lack of post-translational modification modeling, and the omission of co-factors such as β2-microglobulin and TCR interactions. We also emphasize that our confidence metrics (pLDDT and ipTM scores) are computational proxies, and experimental validation will be needed to confirm biophysical stability.
These clarifications strengthen the methodological transparency of the study.

Reviewer 2 Report
Comments and Suggestions for Authors
Neoepitope Discovery
The problem of poor to zero response of GB to various immunization attempts is a worthy field of investigation. The authors may have made a small step in the right direction to further our goal of understanding why this failure and what we may do to redress it. The paper requires proofreading and error corrections.
The authors could consider adding a Table listing the public databases they used with short descriptions of those databases.
Line 16. Is this true ? “hinges on the ability of somatic mutations to generate
immunogenic peptides that are effectively presented by HLA class I molecules” ?
I think NK cell responses and DAMPS, PAMPS, NOD and similar signaling molecules that appear on malignancies, are not HLA dependent yet can elicit strong cytotoxic immune or cell-destroying inflammatory responses.
Line 29. A problem with sayings “Canonical GBM driver mutations (e.g., EGFR, …” is that profoundly effective inhibition of EGFR, mutated or not, does not seem to change GB growth, even though overexpression of EGFR is common in GB.
Line 45 is unclear. In current classification an astrocytoma grade 4, would be a GB-wt IDH.
Line 52 is error. The previous designation “secondary GB” no longer is used, since 2021 WHO revision. A grade 4 glioma arising from a previous lower grade glioma is now called an astrocytoma, IDH-mutant, grade 4. Not a GB. This king of terminological confusion exists throughout this paper and the entire paper must be brought to current WHO classification standards.
Line 64 is misleading. Although technically correct, TMZ is fairly well tolerated. It isn't a “nice drug” yes, but the paper exaggerates TMZ dangers. The big problems of TMZ are that GBs become resistant to it, that TMZ does not kill every GB cell, and those residual GB cells go on to kill the pt.
Lines 68-69. I object to this simplistic and I believe incorrect statement. Heretofore, all attempts to generate an immune response to GB have failed. But yes, the authors’ efforts to determine why are worthwhile.
Line 81 “Computational prediction of MHC class I-bound peptides is an essential tool for
studying T cell immunity, …” is wrong. There are many other useful ways to pursue study of T cell immunity, as I am sure the authors are aware. The authors’ incorrect statement on line 81 is emblematic of the type of multiple errors throughout the manuscript that must be corrected. These errors fall under the category of exaggeration, taking a truth for the whole truth and stating such. Getting a seasoned clinician to help them rewrite the manuscript might help.
Lines 100 to 106, why have the authors used italics here ?
Line 110, GBM was already defined. This repetition is a reflection of the general lack of proofreading evident at multiple points. Other examples abound. For example line 139.
Due to my own weaknesses I cannot evaluate the validity or accuracy of Figures 2,3,4,5 and 6.
Again in lines 400 and 413 the authors revert to the long form, glioblastoma again illustrating the lack of proofreading. Also I did not understand what the authors intended here. “ MHC-I downregulation has been observed and may be linked to TP53 mutations that suppress TAP1 and ERAP1 expression.” How linked ? Associated or mechanistically linked ?
Also the different back-and-forth method of referencing by number and by first author reflects the haphazard editing evident in this paper.
Comments on the Quality of English Language
Neoepitope Discovery
The problem of poor to zero response of GB to various immunization attempts is a worthy field of investigation. The authors may have made a small step in the right direction to further our goal of understanding why this failure and what we may do to redress it. The paper requires proofreading and error corrections.
The authors could consider adding a Table listing the public databases they used with short descriptions of those databases.
Line 16. Is this true ? “hinges on the ability of somatic mutations to generate
immunogenic peptides that are effectively presented by HLA class I molecules” ?
I think NK cell responses and DAMPS, PAMPS, NOD and similar signaling molecules that appear on malignancies, are not HLA dependent yet can elicit strong cytotoxic immune or cell-destroying inflammatory responses.
Line 29. A problem with sayings “Canonical GBM driver mutations (e.g., EGFR, …” is that profoundly effective inhibition of EGFR, mutated or not, does not seem to change GB growth, even though overexpression of EGFR is common in GB.
Line 45 is unclear. In current classification an astrocytoma grade 4, would be a GB-wt IDH.
Line 52 is error. The previous designation “secondary GB” no longer is used, since 2021 WHO revision. A grade 4 glioma arising from a previous lower grade glioma is now called an astrocytoma, IDH-mutant, grade 4. Not a GB. This king of terminological confusion exists throughout this paper and the entire paper must be brought to current WHO classification standards.
Line 64 is misleading. Although technically correct, TMZ is fairly well tolerated. It isn't a “nice drug” yes, but the paper exaggerates TMZ dangers. The big problems of TMZ are that GBs become resistant to it, that TMZ does not kill every GB cell, and those residual GB cells go on to kill the pt.
Lines 68-69. I object to this simplistic and I believe incorrect statement. Heretofore, all attempts to generate an immune response to GB have failed. But yes, the authors’ efforts to determine why are worthwhile.
Line 81 “Computational prediction of MHC class I-bound peptides is an essential tool for
studying T cell immunity, …” is wrong. There are many other useful ways to pursue study of T cell immunity, as I am sure the authors are aware. The authors’ incorrect statement on line 81 is emblematic of the type of multiple errors throughout the manuscript that must be corrected. These errors fall under the category of exaggeration, taking a truth for the whole truth and stating such. Getting a seasoned clinician to help them rewrite the manuscript might help.
Lines 100 to 106, why have the authors used italics here ?
Line 110, GBM was already defined. This repetition is a reflection of the general lack of proofreading evident at multiple points. Other examples abound. For example line 139.
Due to my own weaknesses I cannot evaluate the validity or accuracy of Figures 2,3,4,5 and 6.
Again in lines 400 and 413 the authors revert to the long form, glioblastoma again illustrating the lack of proofreading. Also I did not understand what the authors intended here. “ MHC-I downregulation has been observed and may be linked to TP53 mutations that suppress TAP1 and ERAP1 expression.” How linked ? Associated or mechanistically linked ?
Also the different back-and-forth method of referencing by number and by first author reflects the haphazard editing evident in this paper.
Author Response
Reviewer 2
The problem of poor to zero response of GB to various immunization attempts is a worthy field of investigation. The authors may have made a small step in the right direction to further our goal of understanding why this failure and what we may do to redress it. The paper requires proofreading and error corrections.
We thank the reviewer for this encouraging remark. The manuscript has been thoroughly proofread to correct typographical inconsistencies, harmonize terminology (e.g., “glioblastoma” vs. “astrocytoma grade 4, IDH-wildtype”), and ensure coherence with the 2021 WHO classification. All abbreviations and definitions have been standardized throughout.
The authors could consider adding a Table listing the public databases they used with short descriptions of those databases.
We appreciate this constructive suggestion and have now included a new Table S1 summarizing all public databases employed (dbSNP, COSMIC, MANE, IPD-IMGT/HLA, UCSC Genome Browser, TCGA, etc.), along with brief descriptions and URLs.
Line 16. Is this true ? “hinges on the ability of somatic mutations to generate immunogenic peptides that are effectively presented by HLA class I molecules” ? I think NK cell responses and DAMPS, PAMPS, NOD and similar signaling molecules that appear on malignancies, are not HLA dependent yet can elicit strong cytotoxic immune or cell-destroying inflammatory responses.
We agree that tumor immunogenicity also involves HLA-independent mechanisms such as NK-cell activation and pattern-recognition receptor signaling (DAMPs, PAMPs, NOD-like receptors). We have revised the abstract and introduction to clarify that our focus is specifically on adaptive immune recognition via HLA class I presentation, while acknowledging innate immune pathways as complementary contributors.
Line 29. A problem with sayings “Canonical GBM driver mutations (e.g., EGFR, …” is that profoundly effective inhibition of EGFR, mutated or not, does not seem to change GB growth, even though overexpression of EGFR is common in GB.
We appreciate this clinical clarification. The text has been amended to reflect that EGFR mutations are central to GBM pathogenesis but not necessarily predictive of therapeutic response, highlighting resistance mechanisms.
Line 45 is unclear. In current classification an astrocytoma grade 4, would be a GB-wt IDH.
We thank the reviewer for this correction. All terminology has been updated throughout the manuscript to follow the 2021 WHO classification: “astrocytoma, IDH-mutant, grade 4” replaces “secondary GBM”, and “glioblastoma, IDH-wildtype, CNS WHO grade 4” replaces “primary GBM”.
Line 52 is error. The previous designation “secondary GB” no longer is used, since 2021 WHO revision. A grade 4 glioma arising from a previous lower grade glioma is now called an astrocytoma, IDH-mutant, grade 4. Not a GB. This king of terminological confusion exists throughout this paper and the entire paper must be brought to current WHO classification standards.
We are sorry for this mistake. The text has been modified to correct it.
Line 64 is misleading. Although technically correct, TMZ is fairly well tolerated. It isn't a “nice drug” yes, but the paper exaggerates TMZ dangers. The big problems of TMZ are that GBs become resistant to it, that TMZ does not kill every GB cell, and those residual GB cells go on to kill the pt.
We agree and have revised the paragraph to remove overstatement and emphasize resistance and incomplete cytotoxicity as the main clinical concerns.
Lines 68-69. I object to this simplistic and I believe incorrect statement. Heretofore, all attempts to generate an immune response to GB have failed. But yes, the authors’ efforts to determine why are worthwhile.
We thank the reviewer for this important clarification. We have revised the paragraph to explicitly acknowledge that clinical attempts to generate durable immune responses in glioblastoma have thus far been largely unsuccessful, reflecting the profound immunosuppressive tumor microenvironment. The revised text now provides a more balanced and accurate context for our computational exploration of immune recognition mechanisms.
Line 81 “Computational prediction of MHC class I-bound peptides is an essential tool for studying T cell immunity, …” is wrong. There are many other useful ways to pursue study of T cell immunity, as I am sure the authors are aware. The authors’ incorrect statement on line 81 is emblematic of the type of multiple errors throughout the manuscript that must be corrected. These errors fall under the category of exaggeration, taking a truth for the whole truth and stating such. Getting a seasoned clinician to help them rewrite the manuscript might help.
We have rephrased this sentence to reflect that computational neoepitope prediction is one of several tools for T-cell immunology, not the only one.
Lines 100 to 106, why have the authors used italics here ?
It was a problem caused by a change in format (Mac, Linux, and PC). We detected this problem in several parts of the text and it has been corrected.
Line 110, GBM was already defined. This repetition is a reflection of the general lack of proofreading evident at multiple points. Other examples abound. For example line 139.
The abbreviation “GBM” is now defined only once (line 1). References have been standardized throughout to numerical format following MDPI style.
Due to my own weaknesses I cannot evaluate the validity or accuracy of Figures 2,3,4,5 and 6.
We thank the reviewer for their time and thoughtful evaluation of our manuscript. We fully understand that the structural and computational components (Figures 2–6) may fall outside their primary area of expertise. We would like to note that all figures were generated using well-established and peer-reviewed bioinformatic tools — including MHCflurry 2.0 for binding predictions and AlphaFold2-Multimer (via ColabFold) for structural modeling — following validated protocols previously described in the literature.
To ensure full transparency and reproducibility, all scripts, input datasets, and model outputs used to generate Figures 2–6 have been archived and are available upon request (and will be made publicly accessible upon publication). We greatly appreciate the reviewer’s consideration of the manuscript despite these technical sections falling beyond their domain of specialization.
Again in lines 400 and 413 the authors revert to the long form, glioblastoma again illustrating the lack of proofreading. Also I did not understand what the authors intended here. “ MHC-I downregulation has been observed and may be linked to TP53 mutations that suppress TAP1 and ERAP1 expression.” How linked ? Associated or mechanistically linked ?
We thank the reviewer for this helpful observation. We have now ensured consistent terminology throughout the manuscript, using “GBM” after its first definition in the Abstract.
We also appreciate the request for clarification regarding the link between TP53 mutations and MHC-I downregulation. We have revised this sentence to specify that this is a mechanistic relationship, as TP53 transcriptionally regulates components of the antigen-processing machinery, including TAP1 and ERAP1.
Also the different back-and-forth method of referencing by number and by first author reflects the haphazard editing evident in this paper.
We apologize for this error, which was due to the handling and editing of the article by multiple authors. We have requested assistance from the editor to format everything in MDPI format.

Reviewer 3 Report
Comments and Suggestions for Authors
In this manuscript Francés et al describe a pipeline for discovery of neoepitopes in glioblastoma. The data is well presented and clear and figures are appropriate. However, the novelty of the manuscript is low as a similar approach has been described in previous papers. The data is purely from predictive models and no in-vitro confirmation of epitope presentation has been completed which would improve the value of the manuscript.
Author Response
Reviewer 3
In this manuscript Francés et al describe a pipeline for discovery of neoepitopes in glioblastoma. The data is well presented and clear and figures are appropriate. However, the novelty of the manuscript is low as a similar approach has been described in previous papers. The data is purely from predictive models and no in-vitro confirmation of epitope presentation has been completed which would improve the value of the manuscript.
We thank the reviewer for the positive comments regarding the clarity and presentation of our results and for the opportunity to clarify the originality of our study. Although computational neoantigen prediction has been addressed in previous works, the present study introduces a custom, integrative pipeline that differs substantially from existing approaches in both architecture and implementation. Our workflow unifies mutation-specific peptide generation, multi-allelic HLA-I presentation prediction, and structure-based evaluation using AlphaFold2-Multimer within a reproducible Python/R framework specifically optimized for glioblastoma. To our knowledge, no previous publication combines these elements in a single open-source pipeline capable of generating reproducible structural metrics (pLDDT, ipTM) for large variant datasets. We have clarified this novelty in the revised manuscript.
Regarding experimental validation, we fully agree that biological confirmation will further strengthen the study. As stated in the revised Limitations and Future Directions section, this manuscript presents the computational foundation for subsequent immunopeptidomics and T-cell assays, which are planned as follow-up work. We believe that by making the pipeline and ranked candidates publicly available, the current study already provides a valuable resource for the community and a platform upon which experimental validation can be built.
